Extended Abstract Track

# Neuromorphic Visual Scene Understanding with Resonator Networks (in brief)

**Alpha Renner**                                    ALPREN@INI.UZH.CH
**Giacomo Indiveri**                               GIACOMO@INI.UZH.CH
*Institute of Neuroinformatics, University of Zurich and ETH Zurich, Switzerland*

**Lazar Supic**                              LAZAR.SUPIC@ACCENTURE.COM
**Andreea Danielescu**                ANDREEA.DANIELESCU@ACCENTURE.COM
*Accenture Labs, San Francisco*

**Bruno A. Olshausen**                        BAOLSHAUSEN@BERKELEY.EDU
**Friedrich T. Sommer**                          FSOMMER@BERKELEY.EDU
*Redwood Center for Theoretical Neuroscience, UC Berkeley*

**Yulia Sandamirskaya**                    YULIA.SANDAMIRSKAYA@INTEL.COM
**E. Paxon Frady**                             E.PAXON.FRADY@INTEL.COM
*Intel Neuromorphic Computing Lab*

**Editors:** Sophia Sanborn, Christian Shewmake, Simone Azeglio, Arianna Di Bernardo, Nina Miolane

## Abstract

Inferring the position of objects and their rigid transformations is still an open problem in visual scene understanding. Here we propose a neuromorphic framework that poses scene understanding as a factorization problem and uses a resonator network to extract object identities and their transformations. The framework uses vector binding operations to produce generative image models in which binding acts as the equivariant operation for geometric transformations. A scene can therefore be described as a sum of vector products, which in turn can be efficiently factorized by a resonator network to infer objects and their poses. We also describe a hierarchical resonator network that enables the definition of a partitioned architecture in which vector binding is equivariant for horizontal and vertical translation within one partition, and for rotation and scaling within the other partition. We demonstrate our approach using synthetic scenes composed of simple 2D shapes undergoing rigid geometric transformations and color changes.

## Introduction

Visual scene understanding is a long-standing problem of machine vision and artificial intelligence. It is a notoriously hard – and largely unsolved – computational problem, as it requires searching over a very large space of possible configurations for how objects may be combined along with variations in pose, color, lighting, and other features. It has long been proposed that the brain solves the visual scene understanding problem via "analysis by synthesis" whereby a generative model that holds knowledge of scene components and their combinations is used to infer the components of a scene based on how well they explain the image data (MacKay, 1956; Neisser, 1966; Yuille and Kersten, 2006).

However, the high computational cost associated with these models has prevented their widespread deployment. Recent work has shown that for such types of workloads *neuromorphic computing* can outperform CPU and GPU-based approaches. Specifically, custom spike-based neuromorphic chips (Merolla et al., 2014; Furber et al., 2014; Pei et al., 2019; Davies et al., 2021) can accelerate

Extended Abstract Track

computing times and reduce power consumption thanks to their parallelism, in-memory processing, sparsity, and event-based nature (Indiveri and Liu, 2015; Gallego et al., 2020).

To leverage the advantages of neuromorphic hardware (Kleyko et al., 2021), we base our approach to scene analysis on a programming framework stemming from Cognitive Science now referred to as Vector Symbolic Architectures (VSAs) (Gayler, 2003), or Hyperdimensional Computing (HC) (Kanerva, 2009). Here we leverage results based on VSA/HC to design an algorithm for scene analysis that is optimally suited for neuromorphic hardware implementations. The first development enables us to encode an image in a VSA *hypervector* representation such that binding (vector multiplication) acts as the equivariant operation for specific geometric transformations (Frady et al., 2021), while the second one makes it tractable to infer objects and their transformations via vector factorization (Frady et al., 2020; Kent et al., 2020).

## Representing images in hypervectors

To encode an image as a hypervector, VFA index vectors are created to encode pixel location (Frady et al., 2021). We choose two fixed complex-valued FHRR vectors (Plate, 1995) $\mathbf{h}$ (i.e. $h_j = e^{\iota\phi_j}, \phi_j \sim \mathcal{U}[0, 2\pi]$) and $\mathbf{v}$. A pixel at the Cartesian image coordinates $x$ and $y$ is represented by the index vector $\mathbf{h}^x \odot \mathbf{v}^y$ Further, we form a codebook with random vectors for indexing the three color channels (R/G/B) $\mathbf{G} = [\mathbf{r}, \mathbf{g}, \mathbf{b}] \in \mathbb{C}^{N \times 3}$. The hypervector representation of a color image is then given as:

$$\mathbf{s} = \sum_{x,y,c} Im(x,y,c) \cdot \mathbf{G}_c \odot \mathbf{h}^x \odot \mathbf{v}^y =: \mathbf{\Phi} \, \mathbf{I}. \tag{1}$$

The right hand side of (1) makes explicit that hypervector encoding is linear, with $\mathbf{I} \in \mathbb{R}^{(3P_x P_y)}$ the image reshaped as a vector, and $\mathbf{\Phi} \in \mathbb{C}^{N \times (3P_x P_y)}$ the codebook matrix of hypervectors for each index configuration $\{x, y, c\}$. Conversely, decoding the image from the hypervector uses the conjugate transpose as the linear transform: $\mathbf{I} = \frac{1}{N}\Re(\mathbf{\Phi}^\dagger \mathbf{s})$.

Image encoding with (1) has pivotal properties for the scene factorization algorithm we propose. Most importantly, it ensures that the *equivariant vector operation* for image translation is the binding operation, i.e. $\mathbf{s} \odot \mathbf{h}^{\Delta x} \odot \mathbf{v}^{\Delta y}$ is the VFA representation of the image translated by $\Delta x, \Delta y$, since

$$\mathbf{s} \odot \mathbf{h}^{\Delta x} \odot \mathbf{v}^{\Delta y} = \sum_{x,y} Im(x,y) \cdot \mathbf{h}^{x+\Delta x} \odot \mathbf{v}^{y+\Delta y} = \sum_{x,y} Im(x - \Delta x, y - \Delta y) \cdot \mathbf{h}^x \odot \mathbf{v}^y. \tag{2}$$

Also, note that image translation is well-defined for continuous values of $\Delta x$, $\Delta y$, allowing the recognition of shapes shifted by fractions of a pixel.

## A generative model of scenes using VSA vector operations

To demonstrate scene factorization, we focus on synthetic images of simple visual scenes composed of object templates, in our case letters, that are translated and given one of 7 colors. Multiple objects are independently generated and then added to the scene. The task is to extract from an input image the identities, colors, and locations of the objects.

We can use the VFA framework to build a generative model for such simple synthetic images. The set of letter templates forms a matrix $\mathbf{P} \in \mathbb{R}^{(P_x P_y) \times D}$, where $D = 26$ is the number of different templates. As in (1), templates of the letters can be encoded as hypervectors via $\mathbf{d}_a = \sum_{x,y} \mathbf{P}_a(x, y) \cdot \mathbf{h}^x \odot \mathbf{v}^y$.

The equivariance property of vector binding can be used to color and position a template in the scene. Furthermore, vector superposition is employed for adding different objects. We sample each generative factor of variation $(p_i, x_i, y_i, c_i)$ uniformly. The resulting generative VFA model for

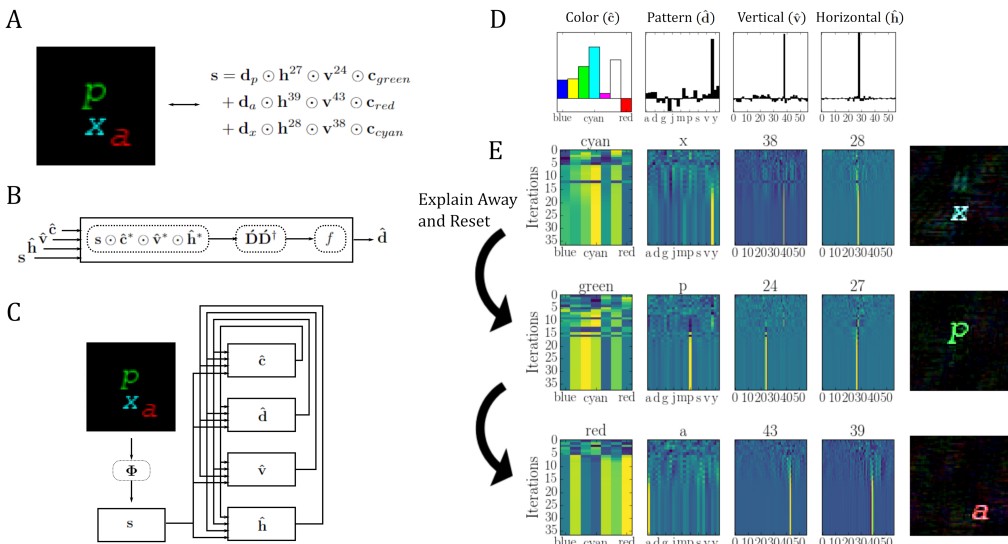

Figure 1: Resonator network for inferring shape, color, and translation.

a synthetic scene composed of $L$ objects is:

$$\mathbf{s} = \sum_{i=1}^{L} \mathbf{d}_{p_i} \odot \mathbf{h}^{x_i} \odot \mathbf{v}^{y_i} \odot \mathbf{c}_{c_i}. \tag{3}$$

## Inference with the resonator network

The generative model (3) allows one to easily compose and render a synthetic scene, but inference in generative models is computationally expensive (Teh et al., 2003) as it involves a combinatorial search across all templates in all possible poses. Conveniently, the VSA formulation (3) permits fast parallel implementations of this search. In particular, each term of the sum in (3) represents an image component formed by the product of hypervectors that encode object class, color, and pose. Thus, inference essentially is the factorization of $\mathbf{s}$ into specific hypervectors that satisfy the optimization constraints.

This kind of vector factorization problem is very common in VSA algorithms, and recurrent resonator networks have been proposed to solve it efficiently (Frady et al., 2020; Kent et al., 2020). A given synthetic pixel image to be analyzed is first transformed by (1) into a hypervector $\mathbf{s}$. The network contains one resonator module that produces an estimate for each factor in the generative model. A resonator network module contains three stages: a VSA binding stage, a linear transform, and a component-wise saturation function or normalization (Fig. 1B).

For inference in (3), the dynamic equations of the resonator network are:

$$\hat{\mathbf{c}}(t+1) = f\left(\acute{\mathbf{C}}\acute{\mathbf{C}}^{\dagger}\left(\mathbf{s} \odot \hat{\mathbf{d}}^{*}(t) \odot \hat{\mathbf{v}}^{*}(t) \odot \hat{\mathbf{h}}^{*}(t)\right)\right),$$

$$\hat{\mathbf{d}}(t+1) = f\left(\acute{\mathbf{D}}\acute{\mathbf{D}}^{\dagger}\left(\mathbf{s} \odot \hat{\mathbf{c}}^{*}(t) \odot \hat{\mathbf{v}}^{*}(t) \odot \hat{\mathbf{h}}^{*}(t)\right)\right),$$

$$\hat{\mathbf{v}}(t+1) = f\left(\mathbf{V}\mathbf{V}^{\dagger}\left(\mathbf{s} \odot \hat{\mathbf{d}}^{*}(t) \odot \hat{\mathbf{c}}^{*}(t) \odot \hat{\mathbf{h}}^{*}(t)\right)\right),$$

$$\hat{\mathbf{h}}(t+1) = f\left(\mathbf{H}\mathbf{H}^{\dagger}\left(\mathbf{s} \odot \hat{\mathbf{d}}^{*}(t) \odot \hat{\mathbf{v}}^{*}(t) \odot \hat{\mathbf{c}}^{*}(t)\right)\right),$$

$$\tag{4}$$

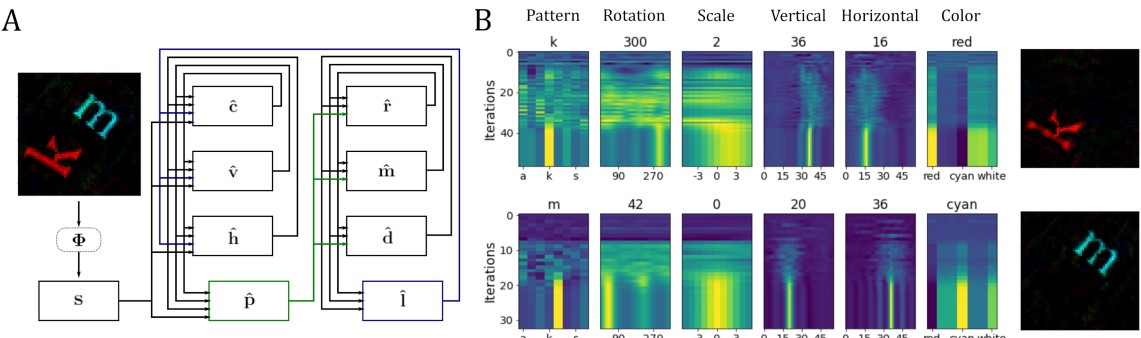

Figure 2: The hierarchical resonator network for inferring rigid transforms.

with $f(x) = x/|x|$ and $\mathbf{V}$, $\mathbf{H}$, $\acute{\mathbf{C}}$, $\acute{\mathbf{D}}$ the codebooks of uncorrelated vectors representing valid constraint vectors. A linear transform of the form $\mathbf{V}\mathbf{V}^{\dagger}$ is essentially a linear auto-associative memory (Kohonen, 1974) that aligns the input to the span of the vectors stored in $\mathbf{V}$. The resonator network (4) solves the inference problem dynamically. Starting from random seeds, in each iteration, a module decodes its own factor from $\mathbf{s}$ by unbinding the estimates from all other modules. Based on vector similarity, the associative memory *cleans up* the decoded vector to resemble one or a superposition of valid codebook vectors. After applying the transfer function $f$, the new estimate is sent to the other modules.

This distributed dynamic process successively improves the joint estimate of all factors (Fig. 1C). Importantly, individual modules do not settle immediately at a single estimate for their factor (like in a Hopfield memory network). In early iteration steps, they produce a superposition of many possible factors, which enables parallel search through the combinatoric solution space. In the later iterations, the interaction between modules narrows the search down to a single estimate of the identity, pose, and color of one scene component, and the network converges to a stable equilibrium (Fig. 1D, E). To analyze other scene components, the previously identified components are subtracted from $\mathbf{s}$, similar to "explaining away" or "deflation."

## Analyzing scenes composed from rigid, non-commutative transforms

The next step toward analyzing realistic scenes is to identify object templates transformed by arbitrary rigid transforms, including translation, rotation, scale, and color. Our approach relies on the fact that scaling and rotation are equivalent to translation in log-polar space. The generative model of an image synthesized from such rigid transforms of shape templates can be written as:

$$\mathbf{s} = \sum_i \mathbf{c}_{c_i} \odot \mathbf{h}^{x_i} \odot \mathbf{v}^{y_i} \odot \boldsymbol{\Lambda}^{-1}(\mathbf{r}^{r_i} \odot \mathbf{m}^{m_i} \odot \mathbf{d}_{p_i}), \qquad (5)$$

where $\boldsymbol{\Lambda}$ is the log-polar transform matrix. For performing inference in this generative model, one can again construct a corresponding resonator network. Describing the six factors in (5), the network consists of six fully connected factor modules that all require coordinate transforms, $\boldsymbol{\Lambda}$ or $\boldsymbol{\Lambda}^{-1}$, in their binding stages. Interestingly, the structure of these equations suggests a partitioned network architecture that avoids redundant coordinate transforms.

The final network consists of two partitions, each fully connected internally: one operating in Cartesian and one in log-polar coordinates. Each partition has an additional module that serves as the communication bridge. Conveniently, the bridge modules have exactly the same internal stages

## Extended Abstract Track

as other resonator modules, a binding stage followed by a linear transform:

$$\hat{\mathbf{l}}(t+1) = \boldsymbol{\Lambda}^{-1}\big(\hat{\mathbf{r}}(t) \odot \hat{\mathbf{m}}(t) \odot \hat{\mathbf{d}}(t)\big), \tag{6}$$

$$\hat{\mathbf{p}}(t+1) = \boldsymbol{\Lambda}\big(\mathbf{s} \odot \hat{\mathbf{c}}^*(t) \odot \hat{\mathbf{h}}^*(t) \odot \hat{\mathbf{v}}^*(t)\big). \tag{7}$$

We describe the schematic of the partitioned resonator network in Fig. 2A. The *log-polar partition*, the right column of modules in Fig. 2A, contains the "top-down" bridge module (6) and modules for inferring identity, rotation and scaling of objects. The *Cartesian partition*, the left column of modules in Fig. 2A, contains the "bottom-up" bridge module (7) and modules for inferring color and translation. We call the architecture in Fig. 2A the *hierarchical resonator network* because the bidirectionally connected partitions assume different hierarchy levels by the definition of Felleman and Van Essen (Felleman and Van Essen, 1991).

A successful example of inference with the hierarchical resonator network is shown in Fig. 2B. The upper row shows a factorization process, revealing the letter "k", and the lower row shows a second factorization process, revealing the letter "m". Note how the estimates of all factors are undecided and blurry in early iteration steps and become sharp quite suddenly during iteration.

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
