# OpenReview forum: "Neuromorphic Visual Scene Understanding with Resonator Networks (in brief)"
_NeurIPS.cc/2022/Workshop/NeurReps — NeurReps 2022 Poster_

### Official Review · Reviewer_2XVg · 2022-10-11
**Low relevance; better fit in a CV venue.**

**Confidence:** 4
**Soundness:** 1
**Presentation:** 1
**Contribution:** 1
**Overall Rating:** 2

**Summary:**

The authors propose an RNN-based architecture for visual scene understanding.
Similar to Capsule networks, their aim is to decouple detecting the object position from detecting its properties such as orientation and color.



**Questions:**

Why did not you consider comparing with Capsule Networks?

**Limitations:**

No mention of limitations.

**Recommended Decision:**

1: Reject

**Relevance:**

1: Not at all relevant

**Strengths And Weaknesses:**

++ Interesting application of RNNs in visual scene understanding.

-- No relevance to the workshop. The authors use the term "geometry" purely in the context geometric transformations in the image space such as translation and rotation. There is no focus on understanding the geometry of neural representations.
The work is a better fit for a CV venue, than to NeurReps.

-- Unclear what the work achieves. The paper ends abruptly without clear results, beyond the activations in Figure 1 and Figure 2.

-- No mention or comparison with Capsule Networks, which aim to achieve more or less the same goal.


**Submission Track:**

Extended Abstract (4 Page)

---

### Official Review · Reviewer_X18Y · 2022-10-14
**Great Presentation. Concise. However, synthetic data should be less behaved when training and evaluating**

**Confidence:** 3
**Soundness:** 3
**Presentation:** 4
**Contribution:** 3
**Overall Rating:** 6

**Summary:**

Understanding object position and pose is still an open-problem in visual scene understanding. The authors propose a neuromorphic framework that uses resonator networks that attempts to extract object identities and their transformations through modelling it as a factorization problem. This work relies on the fact that scaling and rotation are equivalent to translation in log-polar space. Synthetic scenes of 2D shapes undergoing rigid geometric transformations and color changes were used to train and evaluate these networks.

**Questions:**

What happens if your synthetic data is less well-behaved? What happens if the objects are multiple colors? How about scenes with noise?

**Limitations:**

Authors mentioned and show that the dataset is synthetic in nature. A direct statement stating the unknown performance on real data would be good. Would be good to answer a few of the questions in the above box.

**Recommended Decision:**

3: Accept

**Relevance:**

4: Highly relevant

**Strengths And Weaknesses:**

Originality: Clever use of log-polar space, resonator networks, and factorization optimization.

Quality: Technically sound and well-supported. Clearly mentions it uses synthetic, 2D objects.

Clarity: Excellent clarity. Well formatted, figures are clear and writing is clear and concise.

Significance: Good significance. Since the dataset is made of synthetic, well-behaved data, the significance is reduced substantially. How does this model perform under more complicated scenes with backgrounds? How about objects with multiple colors? A good starting point.

**Submission Track:**

Extended Abstract (4 Page)

---

> ### Author Response · Authors · 2022-10-25
> **Thanks**
>
> Hi, thanks for your review. While yes the dataset is "synthetic", this is really just a case that we are fully creating a generative model of a scene. We are extending the system to deal with more natural objects created out of basis vectors rather than templates. The background can be incorporated too, but it is pretty simple to explain away the background. We are also working on extending our framework to more complicated 3D scenes, but this requires novel developments in how we can represent three-dimensional transformations in vector spaces and how to create a 3D generative model. You have to understand that this network does not do any learning or training, it is completely programmed based on the generative model of the scene.
>
> In our generative model, the color of an object is just a particular factor, and the intrinsic shape of an object is not associated with any color. However, this could be slightly molded into the concept of "illumination" color, and then the object also having its own intrinsic colorations. It would be rather simple to incorporate some of these ideas into the generative model, and from there you can reprogram the resonator network to invert this version of the generative model. Creating more complex variations just means incorporating more factors into the generative model, and so if you can build the generative model in the VSA vector space, you can use the resonator network to solve the problem.

---

### Official Review · Reviewer_XkJx · 2022-10-15
**Neuromorphic Visual Scene Understanding**

**Confidence:** 2
**Soundness:** 3
**Presentation:** 3
**Contribution:** 2
**Overall Rating:** 6

**Summary:**

A VSA-based framework, amenable to neuromorphic hardware, is proposed for visual understanding of a scene composed of multiple objects with different configurations of features. A generative model is proposed that can be used to infer components and their features. This model encodes an input image in a factorized fashion to the hyper-vector space where the binding operation is equivalent to the transformations of features. A resonator network is used to perform tractable inference, namely factorize the hyper-vector representation of an image into sources of variations. It is first employed in a simple setting where different letters with arbitrary colors are positioned randomly on an image. Then, a more challenging setting with additional factors of rotation and scale is studied. Thanks to the translational form for these two new factors in log-polar space, the inference is divided into two parts: one part decodes the scale, rotation and identity factors in the log-polar space, while the other decodes the rest in the cartesian space. A communication module (factor) is also introduced which helps unbinding all factors of the current partition when decoding factors in the other partition. In both provided examples, the dynamic process converges to an equilibrium state corresponding to a reasonable disentanglement of rigidly transformed letters in the input image.

**Questions:**

  1. How much is your framework robust to the number of objects in the scene? Suppose an image with dozens of letters. As objects are explained away one at a time, early when several letters are still present in the image, wouldn’t it be difficult for the resonator network to identify an object.
  2. How do you explain sudden shrink in the combinatorial search space and sharp estimates?
  3. In figure 1, part D, what is the y-axis? Also, what does the intensities in part E of the same figure correspond to?
  4. In figure 2, I think that the letter ‘k’ should have a ground-truth rotation around 145 degrees. Then why is 300 written as the estimation?

**Limitations:**

There is no discussion over limitations of the proposed framework.

**Recommended Decision:**

3: Accept

**Relevance:**

4: Highly relevant

**Strengths And Weaknesses:**

* Strengths:
  1. It is well-written. Extending the simple setting to a more challenging one with rotations and scales eases the understanding. The mathematical background on hyper-vector representation has a clear organization.
  2. This work introduces a new useful trick of dividing factors of variations into partitions with each decoding all factors in the same space conditioned on a communication factor from the other partition. One might find this handy for other kinds of transformations, too.

* Weaknesses:
  1. The evaluation is limited as inference is done only for two examples with only few number of letters.
  2. No limitations are provided for the proposed framework


**Submission Track:**

Extended Abstract (4 Page)

---

> ### Author Response · Authors · 2022-10-25
> **Thanks**
>
> Hi, thank you for the review. While we only showed two examples this was merely for space. There is an extended paper on arXiv with the same title if you want further details and discussion of limitations. The network has been tested hundreds of times on many types of scenes, but the analysis of errors is quite complex and depends on a lot of different factors that were too complicated to enumerate. We have a more robust theoretical analysis of errors in our original papers on resonator networks in neural computation.
>
> Questions:
> 1. This really depends on a lot of different factors. Indeed with more objects the problem is harder, but it depends on how large is the scene and how spread out are the objects, how complex are the objects and how much correlations are there between the objects, are the objects overlapping etc. We illustrate the network performing with 3 objects, and it has no problems, and i have done experiments with many more objects. If the problem becomes harder, then this can be compensated by incorporating more neurons into the network.
>
> 2. Well this is the best part of the neural network. The network uses the principle of superposition to search over the combinatoric search space, but once it finds a good solution it rapidly converges. This leads to sharp peaks indicating a strong confidence about one particular combination of factors being present in the scene. The suddenness is just a phenomenon of the postiive feedback in the dynamics once it finds a good combination of factors, it is as if the network has a moment of insight. But this is just how the dynamics empirically behave. This is also partly because the resonator network does not have a traditional lyapunov function and does not simply follow a gradient during this optimization/search.
>
> 3. Yes, the y-axis is merely the readout of the network states. This means just the correlation of the network's vector representations of the objects and the potential outputs. Basically then you accept the largest correlation value of the network as the final output once it converges. The intensities of the heatmap plots in E are showing the exact same correlation function over the iteration dynamics, the figure D is just a bar plot of the final state of the run in panel E. The heatmaps are used to visualize the convergence dynamics and observe the network as it is searching through the state-space.
>
> 4. No, this is a 300 degree rotation, or a -60 degree rotation. Note that 180 degree would be upside down. a 145 degree rotation is would mean the top of the k would be pointing down and right.

---

### Decision · Program_Chairs · 2022-10-21

Accept (Poster)